# Analysis of the release pattern of floral aroma components of *Rhus chinensis* based on HS-SPME-GC-MS technique

Ju Gu[1], Yun Niu[2], Yiting Tang[2], Ping Liu[3], Yandi Wu[1], Zixiang Yang[4]*, Chao Wang[1,2]*

1 Yunnan Province Engineering Research Center for Functional Flower Resources and Industrialization, Southwest Forestry University, Kunming, Yunnan, China, 2 Southwest Landscape Architecture Engineering Research Center of National Forestry and Grassland Administration, Southwest Forestry University, Kunming, Yunnan, China, 3 Yunnan Forestry technological College, Kunming, Yunnan, China, 4 Key Laboratory of Breeding and Utilization of Resource Insects of National Forestry and Grassland Administration, Kunming, Yunnan, China

* yzx1019@163.com (ZY); wchbow@yeah.net (CW)

## Abstract

*Rhus chinensis,* a native plant species of China, possesses significant economic value in the ornamental sector. This study investigates the floral fragrance components and release patterns of *R. chinensis*, thus providing a theoretical foundation for the utilization of its floral fragrance. Headspace-solid phase microextraction (HS-SPME), gas chromatography-mass spectrometry (GC-MS), and chemometrics were used in conjunction with principal component analysis (PCA) and partial least squares-discriminant analysis (PLS-DA) to identify the essential components of the floral aroma during the budding, blooming, and withering stages of *R. chinensis*. The important components of the aroma were also indicated by using the Variable Importance Projections (VIP) and Kruskal-Wallis nonparameters (P). The floral scent components of *R. chinensis* were abundant; 91 and 84 types of floral compounds were found throughout varying flowering seasons and daily patterns, respectively. The primary compounds responsible for flower odors were terpenes, representing over 70% of the floral aroma. Significant fluctuations were observed in the composition of 18 essential scent components and 21 chemicals, with daily variations observed in various flowering stages. The types of floral scent substances continued to rise during the flowering process; however, the relative concentrations of the floral aroma components of *R. chinensis* initially climbed and then fell, reaching 3.60µg/g at the full flowering stage and only 2.40µg/g after the withering stage. In the course of the daily shift, the release amount increased during the day compared to the night, peaking at 4.80µg/g. The substance type reached its greatest point at 12:00, making the circadian rhythm change rule evident. This study provides a reference for the further development and utilization of the flower fragrance of *R. chinensis*.

## Introduction

The floral fragrance of a plant is one of the most crucial indicators of its merit as an ornamental species. These floral scents are constituted by a combination of volatile

**Data availability statement:** All relevant data are within the manuscript and its Supporting Information files.

**Funding:** This study is supported by the Agricultural Joint Special Project of Yunnan Provincial Department of Science and Technology, grant number 202301BD070001 - 087 awarded to CW. The Open Research Program of Key Laboratory of Breeding and Utilization of Resource Insects of National Forestry and Grassland Administration, grant number RIKF202403 and Open project of Key Laboratory for Forest Resources Conservation and Utilization in the Southwest Mountains of China, Ministry of Education, grant number KLESWFU—202005 awarded to CW. The National Natural Science Foundation of China, grant number 32470544 awarded to ZY. The Central Finance Forestry Science and Technology Promotion Demonstration Project, grant number Yun[2024]TG19 awarded to PL.

**Competing interests:** The authors have declared that no competing interests exist.

low-molecular-weight compounds within the flower and are frequently utilized in insect pollination, signaling, as well as in the production of perfumes, fragrances, cosmetics, and pharmaceuticals [1–3]. Furthermore, this floral aroma is widely utilized in horticultural therapy and forest healthcare [4,5].In the context of the remarkable progress and extensive utilization of gas chromatography-mass spectrometry (GC-MS) technology, the integration of GC-MS with multivariate analytical techniques, namely principal component analysis (PCA) and partial least squares-discrimination analysis (PLS-DA), has been prevalently employed in metabolomics analysis as well as in the study of floral fragrance components[6,7]. Currently, the preponderant floral scent compounds within plants can be precisely detected and accurately identified [8].

In accordance with their biosynthetic origins, these compounds are systematically categorized into terpenes, phenolics/phenylpropane, fatty acid derivatives, amino acid derivatives, and certain compounds possessing specific and distinct characteristics [9]. In the plant realm, more than 1700 volatile aroma substances have been accurately identified [10]. Among these, aldehydes, ketones, alcohols, esters, terpenes, alkanes, acids, ethers, and aromatic compounds represent the most prevalent types [11]. Notably, terpenes are exceedingly common and can be ubiquitously detected in nearly all plant floral components [12]. Terpenoids and benzene compounds are predominant in the floral aroma of Magnoliaceae [13]. It has been reported that the floral aroma components of *Osmanthus fragrans* [14], *Lagerstroemia fauriei* [15] and *Panax notoginseng-pinus* [16] are dominated by terpenoids. Moreover, species and flowering stages have significant impacts. The release change rule of plant floral scents is closely related to environmental factors, especially temperature. Additionally, the floral scent components of *Chimonanthus praecox* exhibit a distinct circadian rhythm [17].

*Rhus chinensis* Mill. is a small deciduous tree belonging to the genus Rhus within the Anacardiaceae family. It features panicles, diminutive pure white flowers, purplish-red leaves in autumn, and reddish-orange drupes, thereby possessing a notably high ornamental value [18,19]. It is also a valuable economic tree species in China. Primarily, it serves as a host tree for the Chinese medicinal herb gallnut [20]. *R. chinensis* has been discovered to be one of the principal nectar plants in autumn [21]. Both its female and male flowers possess a high potential for attracting bees [22]. However, as most plants in the Anacardiaceae family possess a distinct odor, the volatile chemicals in their blossoms are frequently overlooked. To date, there has been no report on the study of the fragrance components of *R. chinensis* either domestically or internationally, and systematic research in this regard is lacking. This study employs the combined techniques of headspace-solid phase microextractor (HS-SPME) and gas chromatography-mass spectrometry (GC-MS) to investigate and analyze the floral scent profiles across various flowering stages of *R. chinensis*, encompassing their daily variation patterns. The objective is to elucidate the principal components of *R. chinensis* floral scent and their relative concentrations at distinct phenophases, thereby furnishing references for understanding the metabolic mechanisms of floral aromas and facilitating the effective utilization and development of *R. chinensis* fragrance resources.

## Materials and methods

### Experimental materials

The sampling site of this experiment is an open field cultivation area adjacent to the greenhouse of the Institute of Plateau Forestry, Chinese Academy of Forestry Sciences (located at latitude 25°06′N, longitude 102°76′E, altitude 1982 meters). The region is characterized by a subtropical plateau mountain monsoon climate at low latitudes in the northern hemisphere. It presents a pleasant environment, with no freezing temperatures in winter and no extremely

hot weather in summer. (Ethics Statement: This plant research was conducted in accordance with all relevant ethical guidelines and regulations. Permission has been obtained from the Institute of Plateau Forestry, Chinese Academy of Forestry Sciences.)

Regarding the experimental material, three healthy plants of *R. chinensis* with the same growth vigor were selected. According to three flowering stages (Fig 1): budding stage (flowers are not open, petals are not unfolded), full-flowering stage (more than 80% of the flowers are open, the whole flower is white, the anther filaments are extended, and the petals are fully unfolded) and withering stage (a small number of fallen flowers, flowers slightly wrinkled, petals turned yellow, anther color deepened like light brown) were collected. For the determination of the daily variation rule of floral aroma, flowers of *R. chinensis* in a consistent growth state and at the full-flowering stage were selectively collected from the upper, middle, and lower portions of the plant at 00:00, 06:00, 12:00, and 18:00. Subsequently, they were immediately placed in an ice box and transported back to the laboratory for the determination and analysis of floral aroma components. Three independent replicates were conducted for each stage of each sample.

## HS-SPME analysis

After weighing 1.0 g of the prepared sample into a 20 mL headspace bottle, add 10 μL of the internal standard naphthalene at a concentration of 100 μg/mL and swiftly seal the bottle. Puncture the silicone seal of the headspace cap with a syringe needle and insert the SPME extraction tip (CAR/PDMS, 0.75 μm fiber) into the headspace vial; release the SPME extraction tip, then heated to 80 °C in an electric thermostatic water bath for 50 min. The SPME extraction head was retracted and inserted into the injection port of the GC-MS (7890B-5977B, Agilent, USA), and the sample was injected for 2 min to analyze the SPME extracted sample according to the following conditions.

## GC-MS analysis

With a purity of 99.99% and a flow rate of 1 mL/min, the carrier gas was helium. The inlet temperature was 250°C, with an inlet split ratio of 10:1. The MSD transfer line temperature was 280°C, the MS quadrupole temperature was 150°C, and the ion source temperature was 230°C. The scanning range was 35-450 amu. The ion source was an EI, and the electron bombardment energy was 70eV. The temperature program was 50°C (held for 1 minute), then 180°C at 3°C/min, and ultimately, 250°C at 15°C/min for 3 minutes.

## Data analysis

The mass spectra of each peak were analyzed via gas chromatography-mass spectrometry (GC-MS) in conjunction with the mass spectrometry database and the relative retention time

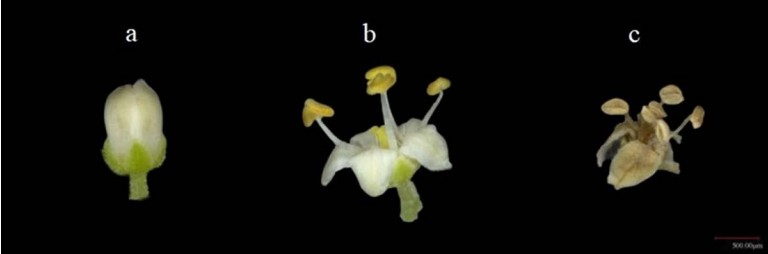

**Fig 1. Morphological characteristics of *R. chinensis* in different flowering stages.** a: Budding stage b: Full flowering stage b: Withering stage.

of each peak. Using the carbon standard method and the same column as well as the same rising and cooling procedures as in GC-MS, a mixed standard of C7-C20 normal alkanes was used as the reference to calculate the linear retention index of various aroma components in *R. chinensis* samples. The results were compared with those of NIST online database. This approach aimed to identify the diverse volatile components in the aroma of *R. chinensis* flowers corresponding to each peak. Quantification was accomplished by comparing the peak area with that of the internal standard to obtain the content of the aroma components in micrograms per gram (μg/g), that is, the content of aroma components = peak area of aroma component substances × content of internal standard/peak area of internal standard. Principal Component Analysis (PCA) and Partial Least Squares Discrimination Analyses (PLS-DA) were conducted using SIMCA 14.1 software to calculate the variable importance in projection (VIP). Additionally, it was combined with SPSS 24.0 software for one-way analysis to calculate the standard error. Differential aroma components were screened with a significance level of $P < 0.05$ and VIP $\geq 1$. TBtools was utilized to draw heat maps, and Origin 2018 software was employed for plotting.

## Results

### Dynamic changes of floral aroma

**Different flowering stages.** Terpenes (29, 69.52%), aldehydes (19, 13.71%), esters (15, 7.80%), alcohols (14, 3.09%), ketones (5, 1.42%), phenols (1, 0.07%), olefins (4, 3.79%), aromatic hydrocarbons (2, 0.20%), acids (1, 0.11%), and other compounds (1, 0.27%) were identified and analyzed(Fig 2). The relative content of floral scent components exhibited a parabolic trend during the flowering process. It was moderate at the budding stage (3.02 μg/g), subsequently reached a maximum at the full flowering stage (3.60 μg/g), and finally decreased from 3.60 μg/g to 2.40 μg/g, resulting in a low content. The quantity of floral scent components progressively increased as the flowers bloomed. Specifically, 69, 83, and 85 floral

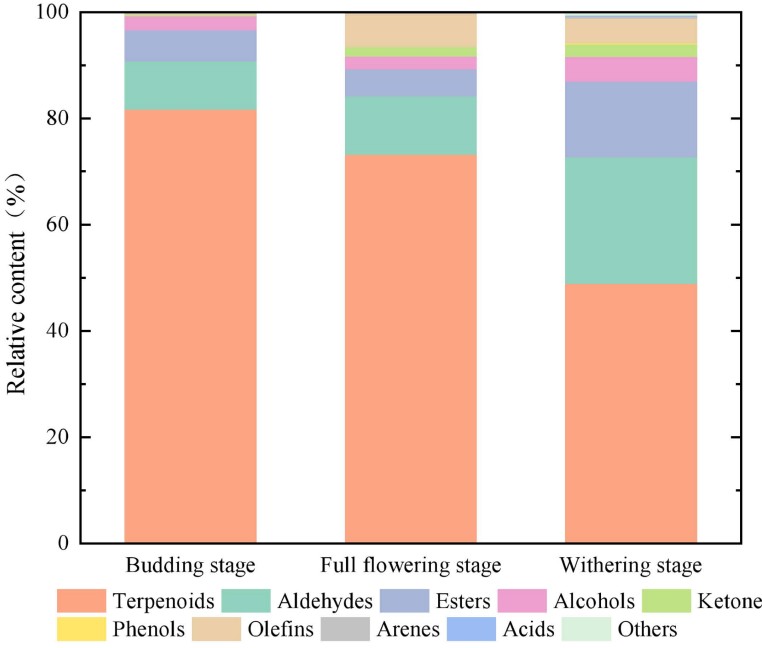

**Fig 2. Percentage of floral fragrance species in different flowering stages of *R. chinensis*.**

aroma compounds were detected at the budding stage, full flowering stage, and withering stage, respectively. There were significant variations in both the types and relative abundances of floral fragrance components among different stages (Table 1).

The terpenoid group exhibited the highest relative abundance among floral aroma components across all three developmental stages. Notably, terpene concentrations demonstrated an initial increase followed by a decline throughout the flowering sequence, with their proportional representation in each stage also diminishing over time (Table 1). During the budding stage, 24 terpenoids were identified, accounting for 81.60% of the total fragrance content. The two predominant constituents are α-Selinene and β-Selinene. Once the plant reaches the full flowering stage, 28 terpenoids, constituting 73.15% of the total, are present. Among them, (E,E)-α-Farnesene has the highest relative quantity, albeit lower than that of α-Selinene and β-Selinene in the budding stage. At the withering stage, the relative amount of terpenes decreased significantly, accounting for 48.88% of the total floral components. Among these, α-Muurolene exhibited the highest relative content, followed by (E)-α-Bergamotene and γ-Cadinene. In comparison with the preceding two stages of the plant, the relative content of these three terpenoids showed a significant increase. Aldehydes, as the second largest group of floral substances in *R. chinensis*, increased significantly in type, relative content and percentage as flowering progressed. The relative content of nonanal was the highest, accounting for about half of the total aldehydes. The release of aldehydes was more significant at the final flowering stage.

**Daily variation.** As the full flowering stage of *R. chinensis* progressed, the total amount of floral scent compounds released exhibited significant daily variations. The dynamic trend of relative content commenced at the lowest value of 2.57 µg/g at 6:00, then gradually rose to 3.60 µg/g at 12:00. Subsequently, it increased again to 4.80 µg/g at 18:00, reaching its maximum. Subsequently, it declined to 3.41 µg/g at 24:00, and its release continued to decrease. During the full flowering stage of *R. chinensis*, 84 floral aroma components were identified (S1 Table). These components included terpenoids (27), aldehydes (17), esters (14), alcohols (13), ketones (4), olefins (3), arenes (3), alkanes (1), acids (1), and others (1) (Table 2).

During the full flowering stage of *R. chinensis*, terpenoids exerted a predominant influence on the daily pattern of variation. The peak release of floral scent was observed at 18:00 in the late afternoon, with a concentration of 3.85 micrograms per gram (µg/g). In contrast, the lowest release of floral scent occurred at 6:00, registering at 1.78 µg/g. The overall content exhibited a similar pattern of initial increase followed by a subsequent decline. At 12:00, (E,E)-α-Farnesene, which is the predominant terpenoid, reached its maximum total release. The changing trend of α-selinene and β-selinene was identical to that of the total content, exhibiting a parabolic trend. Aldehydes constituted the second most significant group of compounds in the daily variation during the full flowering stage of *R. chinensis*, with a maximum release of 0.40 µg/g at 12:00. Among aldehydes, nonanal, the principal floral component, constituted approximately half of the total aldehyde concentration. Subsequent to its release reaching a peak at 6:00 a.m., the relative concentration rose sharply to attain a maximum at 12:00 p.m., after which it commenced a decline once again. The relative quantity of ester compounds has been on the rise in accordance with the daily shift. The increase from 18:00 to 24:00 is conspicuous, with 24:00 witnessing the greatest release (0.31 µg/g), which is dominated by ethyl laurate. he daily variation of olefins was consistent with that of the total floral aroma components, exhibiting a parabolic trend. After reaching the maximum at 18:00 (0.25 µg/g), its relative content decreased significantly. Only three compounds were detected in olefins. However, their relative content was slightly higher than that of esters. The main floral aroma component in olefins was (3E)-4,8-Dimethyl-1,3,7-nonatriene. The remaining compounds were detected only in a small proportion.

**Table 1. Floral aroma components of *R. chinensis* at different flowering stages.**

| Type | RI | Compounds | CAS | Relative content (µg/g) ± SD | | |
|---|---|---|---|---|---|---|
| | | | | Budding stage | Full flowering stage | Withering stage |
| **Terpenoids (29)[2]** | 937 | α-Pinene | 80-56-8 | 0.0718 ± 0.0063b | 0.1150 ± 0.0256a | 0.0546 ± 0.0105b |
| | 979 | β-Pinene | 127-91-3 | —[1] | — | 0.0065 ± 0.0036 |
| | 1026 | D-Limonene | 5989-27-5 | 0.0013 ± 0.0001b | 0.0008 ± 0.0002c | 0.0042 ± 0.0001a |
| | 1049 | Trans-β-Ocimene | 3779-61-1 | — | 0.0020 ± 0.0004a | 0.0007 ± 0.0004b |
| | 1044 | β-Ocimene | 13877-91-3 | 0.0082 ± 0.0013c | 0.0671 ± 0.0089a | 0.0186 ± 0.0003b |
| | 1355 | α-Cubebene | 17699-14-8 | 0.0038 ± 0.0027b | 0.0088 ± 0.0004a | 0.0095 ± 0.0006a |
| | 1386 | β-Bourbonene | 5208-59-3 | 0.0203 ± 0.0010b | 0.0337 ± 0.0043a | 0.0142 ± 0.0014c |
| | 1389 | β-Elemene | 515-13-9 | 0.0281 ± 0.0102b | 0.0843 ± 0.0097a | 0.0215 ± 0.0030b |
| | 1409 | α-Gurjunene | 489-40-7 | 0.0134 ± 0.0015a | 0.0096 ± 0.0030ab | 0.0063 ± 0.0005b |
| | 1423 | β-Caryophyllene | 87-44-5 | 0.1738 ± 0.0072a | 0.0796 ± 0.0045b | 0.0777 ± 0.0077b |
| | 1440 | (+)-Calarene | 17334-55-3 | — | 0.0025 ± 0.0022 | — |
| | 1457 | α-Caryophyllene | 6753-98-6 | 0.1153 ± 0.0118a | 0.0415 ± 0.0017c | 0.0683 ± 0.0016b |
| | 1463 | Alloaromadendrene | 25246-27-9 | 0.0378 ± 0.0096b | 0.0672 ± 0.0077a | 0.0517 ± 0.0021b |
| | 1476 | (4R, 4aS, 6S)-4,4a-Dimethyl-6- (prop-1-en-2-yl) -1,2,3,4,4a, 5,6,7-octahydronaphthalene | 823810-22-6 | 0.0190 ± 0.0069b | 0.0106 ± 0.0014c | 0.0396 ± 0.0007a |
| | 1480 | 2,4,11-Eudesmatriene | 82462-31-5 | 0.1579 ± 0.0192a | 0.0692 ± 0.0095b | 0.0125 ± 0.0037c |
| | 1482 | Germacrene D | 23986-74-5 | 0.0097 ± 0.0006c | 0.0335 ± 0.0085b | 0.0473 ± 0.0005a |
| | 1485 | β-Selinene | 17066-67-0 | 0.6719 ± 0.0130a | 0.3107 ± 0.0175b | 0.0086 ± 0.0021c |
| | 1436 | (E)-α-Bergamotene | 13474-59-4 | 0.0074 ± 0.0003c | 0.0377 ± 0.0037b | 0.2130 ± 0.0144a |
| | 1483 | α-Selinene | 473-13-2 | 0.8506 ± 0.0202a | 0.3766 ± 0.0103b | 0.0099 ± 0.0005c |
| | 1500 | α-Muurolene | 10208-80-7 | 0.0195 ± 0.0014b | 0.0334 ± 0.0052b | 0.2591 ± 0.0098a |
| | 1509 | (E,E)-α-Farnesene | 502-61-4 | 0.0079 ± 0.0013b | 0.9649 ± 0.0181a | 0.0160 ± 0.0021b |
| | 1514 | γ-Cadinene | 39029-41-9 | 0.0236 ± 0.0041b | 0.0269 ± 0.0017b | 0.1709 ± 0.0167a |
| | 1532 | δ-Cadinene | 483-76-1 | 0.2031 ± 0.0086a | 0.1329 ± 0.0154b | 0.0174 ± 0.0010c |
| | 1448 | Cedrene | 11028-42-5 | — | 0.0109 ± 0.0002a | 0.0037 ± 0.0016b |
| | 1535 | Cadinadiene-1,4 | 16728-99-7 | 0.0030 ± 0.0004b | 0.0046 ± 0.0001a | 0.0029 ± 0.0002b |
| | 1434 | β-Copaene | 18252-44-3 | 0.0042 ± 0.0008b | 0.0091 ± 0.0021a | 0.0050 ± 0.0003b |
| | 1538 | α-Calacorene | 21391-99-1 | 0.0084 ± 0.0012a | 0.0105 ± 0.0011a | 0.0178 ± 0.0137a |
| | 1894 | Rimuene | 1686-67-5 | — | 0.0109 ± 0.0038a | 0.0012 ± 0.0007b |
| | 1951 | 13-Isopimaradiene | 1686-56-2 | 0.0024 ± 0.0020b | 0.0787 ± 0.0243a | 0.0129 ± 0.0006b |
| **Aldehydes (19)** | 698 | Valeraldehyde | 110-62-3 | 0.0021 ± 0.0012c | 0.0083 ± 0.0006b | 0.0099 ± 0.0002a |
| | 800 | Hexanal | 66-25-1 | 0.0102 ± 0.0008c | 0.0206 ± 0.0005b | 0.0293 ± 0.0034a |
| | 854 | (E)-2-Hexenal | 6728-26-3 | 0.0043 ± 0.0001c | 0.0161 ± 0.0012b | 0.0214 ± 0.0028a |
| | 900 | Heptaldehyde | 111-71-7 | 0.0068 ± 0.0002b | 0.0157 ± 0.0001a | 0.0163 ± 0.0005a |
| | 914 | Hexa-2,4-dienal | 142-83-6 | — | 0.0002 ± 0.03b | 0.0021 ± 0.0012a |
| | 958 | (E)-2-Heptenal | 18829-55-5 | 0.0052 ± 0.0009b | 0.0076 ± 0.0007b | 0.0190 ± 0.0019a |
| | 962 | Benzaldehyde | 100-52-7 | 0.0028 ± 0.0010c | 0.0100 ± 0.0021b | 0.0487 ± 0.0008a |
| | 1009 | (E,E)-2,4-Heptadienal | 4313-03-5 | — | 0.0039 ± 0.0012b | 0.0143 ± 0.0020a |
| | 1004 | Octanal | 124-13-0 | 0.0173 ± 0.0019b | 0.0256 ± 0.0022a | 0.0157 ± 0.0011b |
| | 1050 | Phenylacetaldehyde | 122-78-1 | 0.0009 ± 0.0005c | 0.0112 ± 0.0015b | 0.0286 ± 0.0028a |
| | 1065 | (E)-Oct-2-enal | 2548-87-0 | 0.0031 ± 0.0001c | 0.0066 ± 0.0006b | 0.0105 ± 0.0004a |
| | 1102 | Nonanal | 124-19-6 | 0.1520 ± 0.0120c | 0.1981 ± 0.0069b | 0.2637 ± 0.0145a |
| | 1156 | (E,Z)-2,6-Nonadienal | 557-48-2 | — | 0.0033 ± 0.0007b | 0.0079 ± 0.0011a |
| | 1162 | (E)-2-Nonenal | 18829-56-6 | 0.0026 ± 0.0005c | 0.0135 ± 0.0017b | 0.0329 ± 0.0071a |
| | 1201 | Decanal | 112-31-2 | 0.0148 ± 0.0046b | 0.0255 ± 0.0012a | 0.0290 ± 0.0015a |
| | 1223 | 2,4-Nonadienal | 6750-03-4 | — | — | 0.0023 ± 0.0004 |
| | 1263 | 2(E)-Decenal | 3913-81-3 | 0.0021 ± 0.0001c | 0.0055 ± 0.0001b | 0.0104 ± 0.0020a |
| | 1306 | Undecanal | 112-44-7 | 0.0498 ± 0.0043a | 0.0324 ± 0.0194b | — |
| | 1715 | Pentadecanal | 2765-11-9 | — | — | 0.0071 ± 0.0018 |

*(Continued)*

**Table 1.** (Continued)

| Type | RI | Compounds | CAS | Relative content (μg/g) ± SD | | |
|------|-----|-----------|-----|------------------------------|---|---|
| | | | | **Budding stage** | **Full flowering stage** | **Withering stage** |
| Esters (15) | 989 | Ethyl caproate | 123-66-0 | 0.0011 ± 0.0002b | 0.0027 ± 0.0012b | 0.0163 ± 0.0014a |
| | 1007 | (Z)-3-Hexen-1-ol acetate | 3681-71-8 | 0.0889 ± 0.0015a | 0.0286 ± 0.0210b | — |
| | 1093 | Ethyl heptanoate | 106-30-9 | 0.0022 ± 0.0005b | 0.0015 ± 0.0001b | 0.0048 ± 0.0005a |
| | 1186 | (E)-3-Hexenyl butyrate | 53398-84-8 | 0.0021 ± 0.0005 | — | — |
| | 1170 | Ethyl benzoate | 93-89-0 | 0.0132 ± 0.0033a | 0.0029 ± 0.0002b | 0.0053 ± 0.0017b |
| | 1193 | Ethyl caprylate | 106-32-1 | 0.0027 ± 0.0017b | 0.0029 ± 0.0008b | 0.0128 ± 0.0019a |
| | 1241 | Cis-3-Hexenyl isovalerate | 35154-45-1 | 0.0204 ± 0.0002a | 0.0053 ± 0.0031b | — |
| | 1247 | Ethyl phenylacetate | 101-97-3 | — | — | 0.0036 ± 0.0007 |
| | 1294 | Ethyl nominate | 123-29-5 | 0.0095 ± 0.0008c | 0.0253 ± 0.0079b | 0.1384 ± 0.0063a |
| | 1392 | Ethyl caprate | 110-38-3 | 0.0009 ± 0.0001b | 0.0054 ± 0.0017ab | 0.0119 ± 0.0059a |
| | 1596 | Ethyl laurate | 106-33-2 | 0.0137 ± 0.0013b | 0.0465 ± 0.0171a | 0.0473 ± 0.0011a |
| | 1686 | Ethyl tridecanoate | 28267-29-0 | 0.0013 ± 0.0001b | 0.0061 ± 0.0033a | 0.0041 ± 0.0002ab |
| | 1797 | Ethyl tetradecanoate | 124-06-1 | 0.0168 ± 0.0030b | 0.0401 ± 0.0211ab | 0.0566 ± 0.0007a |
| | 1996 | Ethyl Palmitate | 628-97-7 | 0.0036 ± 0.0005c | 0.0158 ± 0.0080b | 0.0388 ± 0.0043a |
| | 2194 | Ethyl stearate | 111-61-5 | — | 0.0007 ± 0.0004b | 0.0033 ± 0.0012a |
| Alcohols(14) | 774 | 1-Pentanol | 71-41-0 | 0.0005 ± 0.0003b | 0.0012 ± 0.0004ab | 0.0016 ± 0.0007a |
| | 852 | (E)-3-Hexen-1-ol | 928-97-2 | 0.0366 ± 0.0012a | 0.0240 ± 0.0162b | — |
| | 862 | (E)-2-Hexen-1-ol | 928-95-0 | — | 0.0017 ± 0.0011b | 0.0033 ± 0.0005a |
| | 868 | 1-Hexanol | 111-27-3 | 0.0030 ± 0.0002b | 0.0028 ± 0.0003b | 0.0080 ± 0.0012a |
| | 981 | Oct-1-en-3-ol | 3391-86-4 | 0.0069 ± 0.0009b | 0.0047 ± 0.0001b | 0.0245 ± 0.0068a |
| | 1055 | (E)-2-Octen-1-ol | 18409-17-1 | 0.0015 ± 0.0003c | 0.0023 ± 0.0004b | 0.0097 ± 0.0004a |
| | 1065 | 1-Octanol | 111-87-5 | 0.0106 ± 0.0003b | 0.0125 ± 0.0009ab | 0.0143 ± 0.0028a |
| | 1071 | (Z)-linalool oxide (furanoid) | 5989-33-3 | — | 0.0011 ± 0.0007b | 0.0041 ± 0.0010a |
| | 1105 | Linalool | 78-70-6 | 0.0068 ± 0.0002b | 0.0086 ± 0.0006a | 0.0071 ± 0.0001b |
| | 1118 | Phenylethyl Alcohol | 60-12-8 | 0.0033 ± 0.0010b | 0.0032 ± 0.0004b | 0.0176 ± 0.0006a |
| | 1171 | 1-Nonanol | 143-08-8 | 0.0036 ± 0.0011b | 0.0036 ± 0.0003b | 0.0084 ± 0.0020a |
| | 1211 | Grandlure I | 26532-22-9 | — | 0.0077 ± 0.0002a | 0.0062 ± 0.0008b |
| | 1279 | 1-Decanol | 112-30-1 | 0.0018 ± 0.0005a | 0.0017 ± 0.0003a | 0.0015 ± 0.0005a |
| | 1571 | Nerolidol | 7212-44-4 | 0.0063 ± 0.0023b | 0.0219 ± 0.0067a | 0.0038 ± 0.0005b |
| Ketones(5) | 821 | 3-Cyclohepten-1-one | 1121-64-8 | 0.0009 ± 0.0001b | 0.0010 ± 0.0004b | 0.0095 ± 0.0005a |
| | 979 | 1-Octen-3-one | 4312-99-6 | 0.0019 ± 0.0002b | 0.0031 ± 0.0003b | 0.0063 ± 0.0014a |
| | 987 | 6-Methyl-5-heptene-2-one | 110-93-0 | 0.0038 ± 0.0004b | 0.436 ± 0.0323a | 0.0155 ± 0.0035ab |
| | 1107 | 6-Methyl-3,5-Heptadien-2-One | 1604-28-0 | — | — | 0.0045 ± 0.0006 |
| | 1147 | 2,6,6-Trimethyl-2-cyclohexene-1,4-dione | 1125-21-9 | — | 0.0168 ± 0.0002b | 0.0213 ± 0.0038a |
| Phenols(1) | 1360 | Eugenol | 97-53-0 | — | 0.0011 ± 0.0006b | 0.0057 ± 0.0012a |
| Olefins(4) | 880 | 1.3.5-Octatriene | 3806-77-9 | — | 0.0032 ± 0.0008b | 0.0134 ± 0.0031a |
| | 1053 | 2,6-Octadiene, 2,6-dimethyl- | 2792-39-4 | — | 0.0028 ± 0.0008b | 0.0045 ± 0.0006a |
| | 1117 | (E)-4,8-Dimethylnona-1,3,7-triene | 19945-61-0 | 0.0051 ± 0.0006c | 0.2196 ± 0.0198a | 0.0892 ± 0.0047b |
| | 1821 | (E)-5-Octadecene | 7206-21-5 | — | — | 0.0045 ± 0.0006 |
| Arenes(1) | 898 | Styrene | 100-42-5 | — | — | 0.0032 ± 0.0006 |
| Acids(1) | 1189 | Thianaphthene | 95-15-8 | 0.0064 ± 0.0011a | 0.0042 ± 0.0001b | 0.0045 ± 0.0008b |
| | 1280 | Nonanoic acid | 112-05-0 | 0.0035 ± 0.0005b | 0.0015 ± 0.0009c | 0.0052 ± 0.0009a |
| Others(1) | 994 | 2-Amylfuran | 3777-69-3 | 0.0026 ± 0.0006b | 0.0049 ± 0.0009b | 0.0165 ± 0.0033a |
| Total | | 91 | | 3.0179 ± 0.0189b | 3.6213 ± 0.0991a | 2.3987 ± 0.0260c |

[1]Not detected or nonexistent.

[2]Number of compounds. Lowercase letters indicate significant differences at the $p \leq 0.05$ level.

**Table 2. Diurnal variation classification statistics of flower fragrance components of *R. chinensis*.**

| Compounds | Relative content (μg/g) ± SD | | | |
|---|---|---|---|---|
| | 06:00 | 12:00 | 18:00 | 24:00 |
| Terpenoids(27) | 1.7818 ± 0.00067d(26)[1] | 2.6223 ± 0.1259b(27) | 3.8502 ± 0.0384a(26) | 2.3436 ± 0.0094c(27) |
| Aldehydes(17) | 0.2724 ± 0.0025d(15) | 0.4042 ± 0.0035a(17) | 0.3379 ± 0.0207c(15) | 0.3805 ± 0.0031b(16) |
| Esters(14) | 0.1474 ± 0.0106c(13) | 0.1836 ± 0.0456b(14) | 0.2195 ± 0.0021b(13) | 0.3082 ± 0.0071a(13) |
| Alcohols(13) | 0.1155 ± 0.0210b(12) | 0.0720 ± 0.0042c(13) | 0.0754 ± 0.0002c(13) | 0.1536 ± 0.0055a(13) |
| Ketones(4) | 0.0132 ± 0.0022b(3) | 0.0645 ± 0.0327a(4) | 0.0199 ± 0.0005b(4) | 0.0279 ± 0.0017b(4) |
| Olefins(3) | 0.1905 ± 0.0085c(3) | 0.2251 ± 0.0199b(3) | 0.2527 ± 0.0067a(2) | 0.1691 ± 0.0052c(3) |
| Arenes(3) | 0.0418 ± 0.0036a(3) | 0.0098 ± 0.0050c(2) | 0.0203 ± 0.0011b(2) | 0.0097 ± 0.0004c(3) |
| Alkanes(1) | 0.0022 ± 0.0009d(1) | 0.0091 ± 0.0021b(1) | 0.0176 ± 0.0013a(1) | 0.0048 ± 0.0001c(1) |
| Acids(1) | 0.0037 ± 0.0007a(1) | 0.0015 ± 0.0008b(1) | 0.0011 ± 0.0004b(1) | 0.0042 ± 0.0003a(1) |
| Others(1) | 0.0024 ± 0.0004c(1) | 0.0049 ± 0.0009a(1) | 0.0034 ± 0.0004b(1) | 0.0039 ± 0.0003b(1) |
| Total(84) | 2.5710 ± 0.0100d(78) | 3.5970 ± 0.0935b(83) | 4.7981 ± 0.0146a(78) | 3.4056 ± 0.0187c(82) |

[1]Number of compounds. Lowercase letters indicate significant differences at the p ≤ 0.05 level.

## Chemometric analysis of floral aroma components

Principal Component Analysis (PCA) represents a multivariate statistical analysis approach wherein several variables are subjected to linear transformation to select a smaller number of meaningful variables. It can provide a preliminary assessment of the overall metabolite differences between samples within groups as well as the degree of variability within groups. When creating the components, Partial Least Squares Discrimination Analysis (PLS-DA) is a supervised discriminant analysis approach [23], a multivariate statistical analysis method that takes into consideration the class membership information supplied by the auxiliary matrices in the form of codes [24]. Compared with Principal Component Analysis (PCA), Partial Least Squares Discrimination Analysis (PLS-DA) exhibits a higher degree of separation. It can analyze metabolites based on predefined categories (Y variables) to maximize the differences between groups. A more stable model is characterized by a closer approximation of R2X to 1. Moreover, Q2 being greater than 0.5 indicates a high prediction rate, suggesting that the model is well-stabilized and highly predictable [25].

As depicted in Fig 3, significant disparities were observed in the floral scent composition of *R. chinensis* at different stages. After fitting, in the PCA model (Fig 3(a)), two principal components, namely PC1 accounting for 61.2% and PC2 accounting for 27.4%, with a cumulative contribution of 88.6% were identified for various flowering stages. Similarly, in the PLS-DA model (Fig 3(b)), two principal components, PC1 accounting for 61.4% and PC2 accounting for 35.1%, with a cumulative contribution of 96.5% were found. The model prediction index (Q2) was 0.997, the dependent variable fit index (R2y) was 0.999, and the independent variable fit index (R2x) was 0.964. The two principle components in the PCA (Fig 3(c)) and PLS-DA (Fig 3(d)) models for daily variation, PC, were PC1 = 42.9% and PC2 = 26.7%, with a cumulative contribution of 69.6%; the two main components in the PLS-DA model, with a total contribution of 84%, are PC1 = 66.5% and PC2 = 17.5%. The model has strong stability and predictability, as evidenced by the fitting indices of the independent variable (R2x) of 0.979, the dependent variable (R2y) of 0.997, and the model prediction index (Q2) of 0.993.

The results of the analyses comprehensively mirrored the original information regarding the three flowering stages and daily variations in the flower fragrance of *R. chinensis*. The classification algorithm effect of the two datasets was excellent. It enabled the separation of sample points without any cross-cutting among the samples. The samples within the same flowering

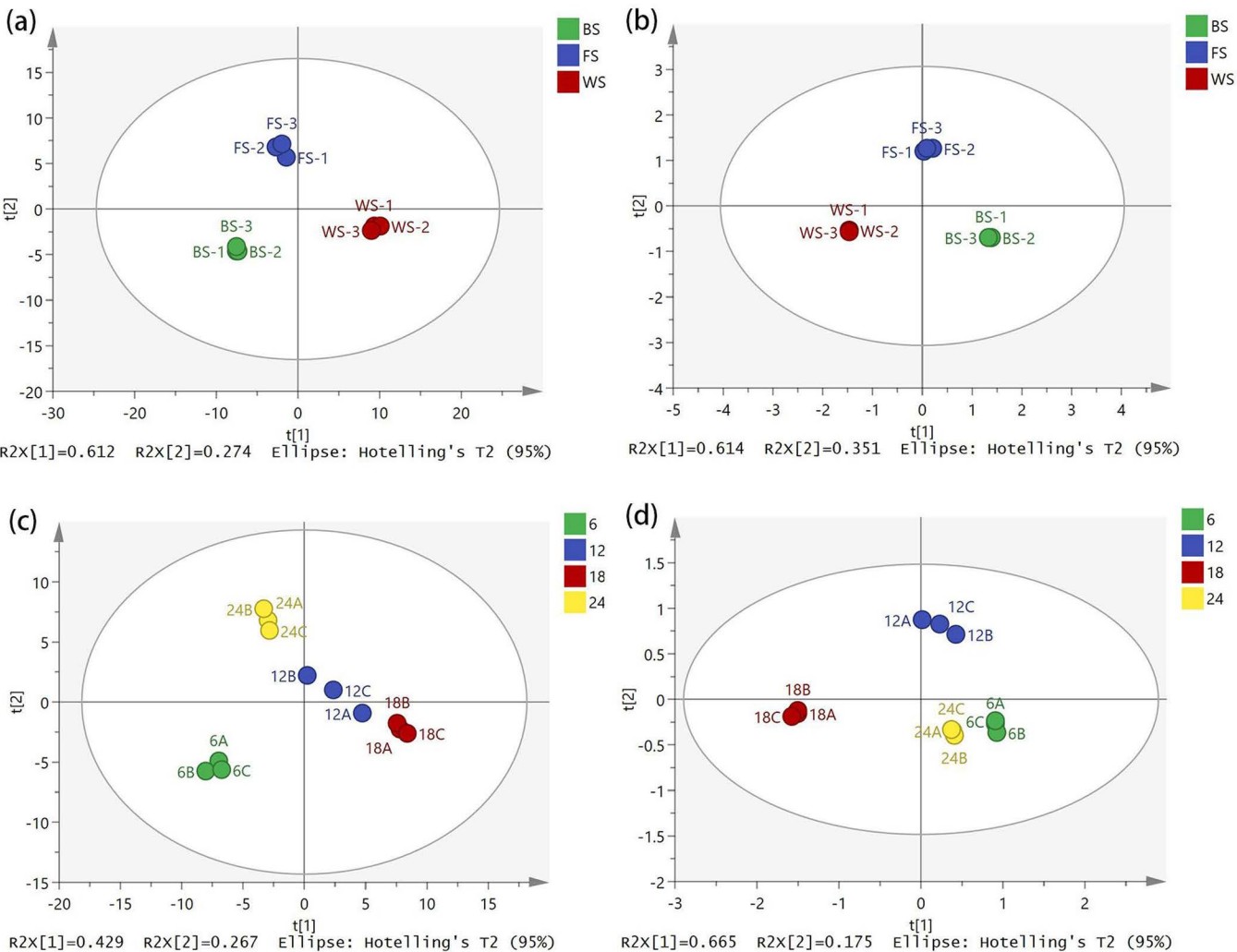

**Fig 3. Model analysis of floral aroma components in *R. chinensis*: (a) PCA score plot of different flowering stages; (b) PLS-DA score plot of different flowering stages; (c) PCA score plot of Diurnal Variation; (d) PLS-DA score plot of Diurnal Variation.**

stage demonstrated excellent repeatability, indicating that variations are present in the scent components of *R. chinensis* flowers across distinct flowering stages and temporal intervals. Upon conducting 200 permutation tests (Fig 4), the results showed that: R2 = (0.0, 0.189), Q2 = (0.0, -0.302) for the various floral stages of *R. chinensis*. The intercept values of the predicted points of the floral aroma components of *R. chinensis* for the three floral stages and daily variations were less than those of the original model (Q2 < R2). Moreover, the point of intersection of the Q2 regression line with the vertical axis was less than 0, signifying that the model was validated and did not exhibit overfitting. It was considered that the results could be used for the identification and analysis of floral and daily variations in the aroma of *R. chinensis* flowers.

## Determination of crucial components

**Different flowering stages.** The variable importance in projection (VIP) for variable significance can serve as an indicator to illustrate how diverse variables contribute to the overall categorization [26]. The higher the VIP values of the groups are, the more significant

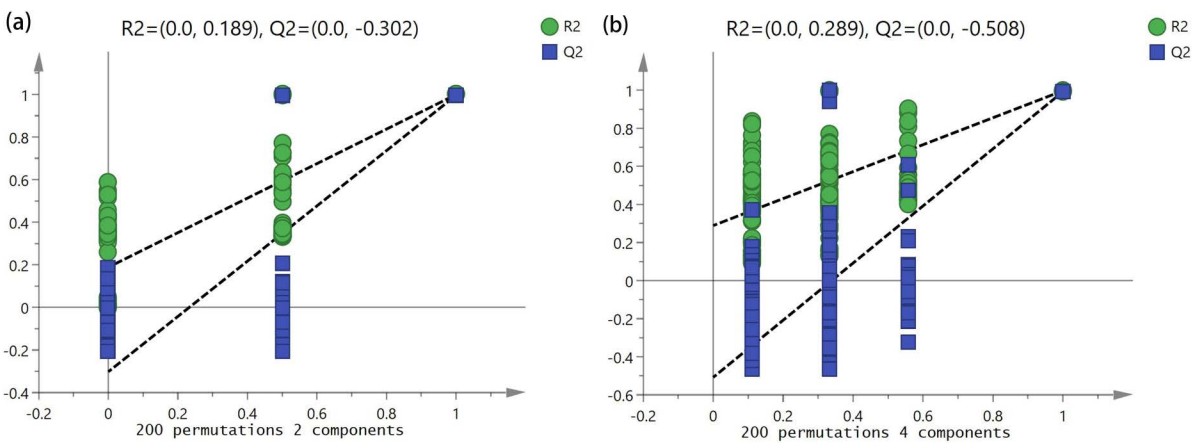

**Fig 4. Permutation test of PLS-DA mode: (a) Different flowering stages; (b) Diurnal Variation.**

the differences in the variables between them become. When the VIP value exceeds 1.00, the corresponding variable can be regarded as the key variable of the discriminant model. The selection criterion of p-value < 0.05 is employed. Compounds with VIP > 1.00 are further examined in an endeavor to enhance the accuracy of the analysis. The compounds with VIP > 1.00 were subjected to re-examination. The selection criterion was set as p < 0.05 to enhance the analytical precision. Eighteen key compounds (Table 3) were obtained for distinguishing different flowering stages of *R. chinensis*, namely (E,E)-α-Farnesene, α-Selinene, β-Selinene, (E)-4,8-Dimethylnona-1,3,7-triene, α-Muurolene, (E)-α-Bergamotene, γ-Cadinene, δ-Cadinene, Ethyl stearate, 2,4,11-Eudesmatriene, 13-Isopimaradiene, β-Caryophyllene, β-Elemene, α-Caryophyllene, Nonanal, β-Ocimene, (Z)-3-Hexen-1-ol acetate, and α-Pinene.

In order to visualize the alterations in floral aroma compounds among different stages of *R. chinensis*, a clustered heat map analysis was conducted on the distribution of the 18 key components in various flowering stages, as depicted in Fig 5. The 18 key components were uniformly and abundantly distributed in different locations, enabling them to completely segregate the three flowering stages. The outcomes demonstrated that the various flowering stages differed significantly from one another. Among them, the bloom stage had larger concentrations of δ-Cadinene, (Z)-3-Hexen-1-ol acetate, α-Selinene, β-Selinene, β-Caryophyllene, and α-Caryophyllene than the budding and withering stages. The content of (E,E)-α-Farnesene was higher at the withering stage.

**Daily variation.** By employing the PLS-DA model to rank the importance of variable projections, compounds with VIP > 1.00 in the daily variation were identified. A total of 21 differential compounds (Table 4), including (E,E)-α-Farnesene, α-Selinene, β-Selinene, Ethyl tetradecanoate, Ethyl laurate, (E)-2-Hexenal, Nonanal, 13-Isopimaradiene, α-Pinene, (E)-4,8-Dimethylnona-1,3,7-triene, Nerolidol, δ-Cadinene, β-Elemene, Methylheptenone, β-Caryophyllene, β-Ocimene, Phenylallene, 2,4,11-Eudesmatriene, α-Calacorene, Ethyl Palmitate, and Alloaromadendrene, were screened by combining the criterion of p-value < 0.05.

The results of an analysis utilizing a clustered heat map to examine the distribution of the 21 key components of *R. chinensis* at different blooming stages are presented in Fig 6. The findings revealed that the various times of the day exhibited significant variations from one another. Moreover, the 21 key components enabled complete segregation of the different periods. Among these components, the concentration of floral compounds, namely

Table 3. Characteristics of the 18 key components in the PLS-DA model.

| NO | CAS | Compounds | VIP | P |
|---|---|---|---|---|
| 1 | 502-61-4 | (E,E)-α-Farnesene | 5.02428 | 0 |
| 2 | 473-13-2 | α-Selinene | 3.34948 | 0 |
| 3 | 17066-67-0 | β-Selinene | 2.9666 | 0 |
| 4 | 19945-61-0 | (E)-4,8-Dimethylnona-1,3,7-triene | 2.16802 | 0 |
| 5 | 10208-80-7 | α-Muurolene | 2.00775 | 0 |
| 6 | 13474-59-4 | (E)-α-Bergamotene | 1.76641 | 0 |
| 7 | 39029-41-9 | γ-Cadinene | 1.58282 | 0 |
| 8 | 483-76-1 | δ-Cadinene | 1.56764 | 0 |
| 9 | 111-61-5 | Ethyl stearate | 1.41305 | 0.002 |
| 10 | 82462-31-5 | 2,4,11-Eudesmatriene | 1.39592 | 0 |
| 11 | 1686-56-2 | 13-Isopimaradiene | 1.34023 | 0.001 |
| 12 | 87-44-5 | β-Caryophyllene | 1.32174 | 0 |
| 13 | 515-13-9 | β-Elemene | 1.23423 | 0 |
| 14 | 6753-98-6 | α-Caryophyllene | 1.20301 | 0 |
| 15 | 124-19-6 | Nonanal | 1.20176 | 0 |
| 16 | 13877-91-3 | β-Ocimene | 1.1916 | 0 |
| 17 | 3681-71-8 | (Z)-3-Hexen-1-ol acetate | 1.10072 | 0 |
| 18 | 80-56-8 | α-Pinene | 1.09625 | 0.01 |

(E,E)-α-Farnesene, α-Selinene, β-Selinene, Nonanal, and α-Pinene, was lower during the early morning and wee hours compared to that in daylight at 12:00 and 18:00. The level of (E,E)-α-Farnesene at 12:00 was significantly higher than that at any other time point. Hierarchical cluster analysis demonstrated that the distribution of the differential substance content at four distinct time points was divided into two groups: one group was formed between 6:00 and 24:00, while the other group was established between 12:00 and 18:00.

## Discussion

The compositions and concentrations of floral volatiles play a decisive role in determining the variations of plant floral fragrances [27]. It has been demonstrated that as flowers open and progress through development, the types of fragrance compounds they generate and their relative abundances exhibit significant variations during different developmental stages [28]. In this study, a total of 91 distinct floral fragrance compounds were detected and identified in different flowering stages of *R. chinensis*. The types and concentrations of volatile components exhibited certain variations. A parabolic trend was observed in the variation of relative content from the budding stage to the withering stage, with the release reaching its peak at the full-flowering stage. This is consistent with previous research findings on *Michelia crassipes* [29], *Jasminum sambac* [30], *Angraecum sesquipedale* [31] and other plants. A comparison among the various flowering stages revealed that the floral scent composition was at its peak during the withering stage, where more specific aldehydes, esters and other compounds could be detected. The relative content of major terpenoids in the floral scent composition was relatively high during the budding stage. The total release of floral scent increased during the full flowering stage, and (E,E)-α-Farnesene was present in substantial amounts.

The floral components of *R. chinensis* possess significant utilization value. Terpenoids constituted the type of compounds present in the largest quantity and with relatively high contents in the floral scent of *R. chinensis*. The majority of them were identified in all

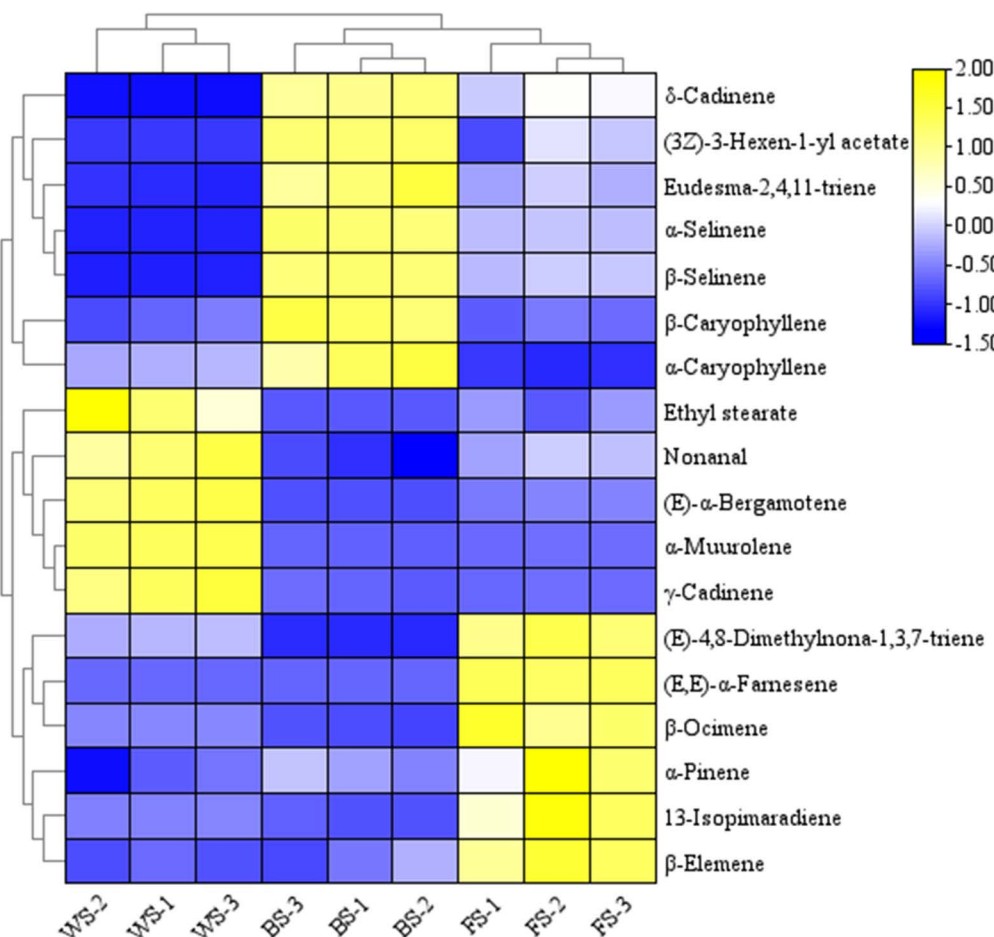

**Fig 5. Heat map of the content distribution of 18 crucial components in different flowering stages.**

three flowering stages. Certain terpenoids, for instance, (E,E)-α-Farnesene, α-Selinene, β-Selinene, and others, which are abundant in a specific stage, can be extracted and utilized, and hold high development potential. One of the key constituents of rose fragrance was α-Caryophyllene [32], which could be employed to produce Xinjiang Damask rose essential oil [33]. The compounds of *R. chinensis*, including α-Selinene, β-Selinene, δ-Cadinene, β-Caryophyllene, and α-Caryophyllene, were predominantly present in the budding stage. However, as the plants entered the flowering stage, the release of these compounds diminished. The predominant volatile constituent in the petals of 19 intergroup hybrids of the genus Paeonia L. was β-caryophyllene, which exhibited a woody odor [34]. The second major group of compounds contributing to the floral scent of *R. chinensis* was aldehydes, with nonanal being the main constituent. Its relative amount progressively increased as the flowers opened and attained its peak concentration after the flowering stage. Nonanal can be utilized in the production of scents such as rose, orange flower, scented violet, and incense. It exhibits a rich, oily odor with a sweet orange undertone [35].

There exist pronounced circadian rhythm fluctuations in the release pattern of plant floral fragrance. These fluctuations are highly correlated with environmental conditions, with temperature being of particular significance [36,37]. The study outcomes unveiled

**Table 4. Characteristics of the 21 crucial components in the PLS-DA model.**

| NO | CAS | Compounds | VIP | P |
|---|---|---|---|---|
| 1 | 502-61-4 | (E,E)-α-Farnesene | 3.44888 | 0 |
| 2 | 473-13-2 | α-Selinene | 2.87893 | 0 |
| 3 | 17066-67-0 | β-Selinene | 2.44923 | 0 |
| 4 | 124-06-1 | Ethyl tetradecanoate | 2.08373 | 0 |
| 5 | 106-33-2 | Ethyl laurate | 1.8879 | 0 |
| 6 | 6728-26-3 | (E)-2-Hexenal | 1.81969 | 0 |
| 7 | 124-19-6 | Nonanal | 1.66499 | 0 |
| 8 | 1686-56-2 | 13-Isopimaradiene | 1.58441 | 0.002 |
| 9 | 80-56-8 | α-Pinene | 1.54142 | 0 |
| 10 | 19945-61-0 | (E)-4,8-Dimethylnona-1,3,7-triene | 1.53957 | 0 |
| 11 | 7212-44-4 | Nerolidol | 1.46406 | 0 |
| 12 | 483-76-1 | δ-Cadinene | 1.40859 | 0 |
| 13 | 515-13-9 | β-Elemene | 1.38878 | 0 |
| 14 | 409-02-9 | Methylheptenone | 1.36598 | 0.048 |
| 15 | 87-44-5 | β-Caryophyllene | 1.3189 | 0 |
| 16 | 13877-91-3 | β-Ocimene | 1.22741 | 0.002 |
| 17 | 2327-99-3 | Phenylallene | 1.19572 | 0 |
| 18 | 82462-31-5 | 2,4,11-Eudesmatriene | 1.18261 | 0 |
| 19 | 21391-99-1 | α-Calacorene | 1.16744 | 0 |
| 20 | 628-97-7 | Ethyl Palmitate | 1.09389 | 0.001 |
| 21 | 25246-27-9 | Alloaromadendrene | 1.08201 | 0 |

the diurnal variation pattern of *R. chinensis*. The proportionate content of floral scent constituents initially ascended and then descended over time. Moreover, the daytime levels were markedly higher than those at nighttime. This pattern of change was consistent with that observed in *Dendrobium chrysotoxum* [38] and *Chionanthus retusus* [39]. A distinct pattern of circadian rhythm shift was observed in the groups formed by cluster analysis: 12:00 and 18:00 during the day and 6:00 and midnight. *R. chinensis* possessed the highest number of types of floral scent compounds at noon. The amount of floral aroma components released increased from 12:00 to 18:00, and the relative content reached its peak at 18:00. This may be due to the high temperature and intense light, which facilitate the release. The study findings support the diurnal fluctuations observed in *Malus spectabilis* [40] and *Lilium*'Siberia' [41].

In this study, among the 21 key floral components in daily fluctuation and the 18 key floral components identified at various flowering stages, the predominant ones were terpenoids. Terpenoids not only possess significant biological functions but also are intricately involved in physiological processes such as plant growth and development. For instance, (E,E)-α-farnesene exerts a significant influence on the insect resistance of many plants [42]. α-Selinene and β-Selinene display remarkable broad-spectrum antibacterial activity. Regarding the hypoxic injury of H9c2 cells, cadinene demonstrates certain protective properties [43]. Regarding the hypoxic injury of H9c2 cells, cadinene exhibits certain protective properties [44]. Elemene shows remarkably strong antitumor properties [45,46]. Nerolidol can be utilized to prepare flavors reminiscent of rose and syringa. It exerts an aroma-fixing influence, demonstrates good persistence, and exhibits certain coordinating capabilities [47]. These compounds possess significant aromatic characteristics that are conducive to human health and emotional well-being.

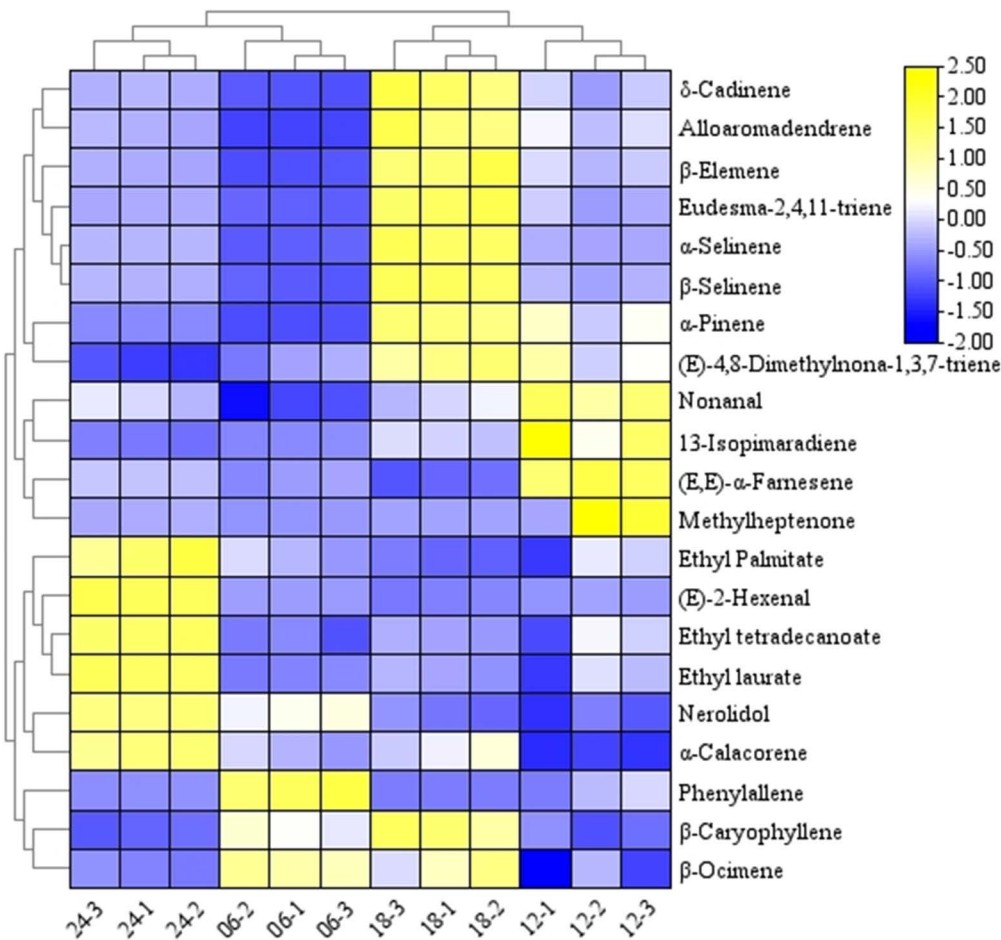

**Fig 6. Diurnal variation of 21 key components of the content distribution heat map.**

## Conclusion

The findings of this study demonstrated that the floral substances of *R. chinensis* were primarily terpenoids. As the flowering process advanced, the release of floral aroma substances exhibited a parabolic trend, reaching its maximum at the full flowering stage. There was a distinct pattern of variation in the diurnal rhythm. Due to environmental conditions and other influences, the daytime release of floral compounds of *R. chinensis* was significantly greater than that at night. Cluster analysis demonstrated that the principal constituents of floral fragrance showed variations in accordance with daily variation patterns and flowering stages. Furthermore, principal component analysis (PCA) revealed that (E,E)-α-farnesene, α-selinene, and β-selinene were the predominant substances differentiating the four varieties of *R. chinensis*. The volatile components of the floral fragrance of *R. chinensis* are abundant and hold significant value in economic, ecological, and medicinal contexts.

## Supporting information

**S1 Fig. Total ion chromatogram of floral fragrance components of *R. chinensis* at different flowering stages.**
(TIF)

**S2 Fig. Total ion chromatogram of floral fragrance components of *R. chinensis* at daily Variation.**
(TIF)

**S1 Table. Floral aroma components of *R. chinensis* at daily Variation.**
(XLSX)

## Author contributions

**Conceptualization:** Ju Gu, Zixiang Yang, Chao Wang.

**Data curation:** Ju Gu.

**Formal analysis:** Ju Gu.

**Funding acquisition:** Ping Liu, Zixiang Yang, Chao Wang.

**Investigation:** Yiting Tang, Ping Liu.

**Methodology:** Ju Gu.

**Project administration:** Chao Wang.

**Resources:** Zixiang Yang, Chao Wang.

**Software:** Yun Niu.

**Supervision:** Zixiang Yang, Chao Wang.

**Validation:** Ju Gu, Yiting Tang, Yandi Wu.

**Visualization:** Yandi Wu.

**Writing – original draft:** Ju Gu.

**Writing – review & editing:** Ju Gu.

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
