## [Decision Letter · Decision Letter 0]

6 Dec 2024

PONE-D-24-52009Analysis of the release pattern of floral aroma coyunmponents of Rhus chinensis based on HS-SPME-GC-MS techniquePLOS ONE

Dear Dr. Wang,

Thank you for submitting your manuscript to PLOS ONE. After careful consideration, we feel that it has merit but does not fully meet PLOS ONE’s publication criteria as it currently stands. Therefore, we invite you to submit a revised version of the manuscript that addresses the points raised during the review process.

**ACADEMIC EDITOR: ** Dear Dr. Wang, thank you for submitting your manuscript for consideration in Plos One. I have received the reviewers' recommendations regarding your work. Please provide a new, updated and corrected version of your manuscript and a letter with all the reviewers' recommendations point by point for each issue. 

We look forward to receiving your revised manuscript.

Kind regards,

Mozaniel Santana de Oliveira, Ph.D

Academic Editor

PLOS ONE

Journal Requirements: When submitting your revision, we need you to address these additional requirements. 1. Please ensure that your manuscript meets PLOS ONE's style requirements, including those for file naming. The PLOS ONE style templates can be found at https://journals.plos.org/plosone/s/file?id=wjVg/PLOSOne_formatting_sample_main_body.pdf and https://journals.plos.org/plosone/s/file?id=ba62/PLOSOne_formatting_sample_title_authors_affiliations.pdf 2. Thank you for stating the following financial disclosure: "This study was supported by Agricultural Joint Special Project of Yunnan Provincial Department of Science and Technology, grant number 202301BD070001-087; Open Research Program of Key Laboratory of Breeding and Utilization of Resource Insects of National Forestry and Grassland Administration, grant number XXX; Central Finance Forestry Science and Technology Promotion Demonstration Project, grant number Yun[2024]TG19 , and Open project of Key Laboratory for Forest Resources Conservation and Utilization in the Southwest Mountains of China, Ministry of Education, grant number KLESWFU——202005." Please state what role the funders took in the study.  If the funders had no role, please state: ""The funders had no role in study design, data collection and analysis, decision to publish, or preparation of the manuscript."" If this statement is not correct you must amend it as needed. Please include this amended Role of Funder statement in your cover letter; we will change the online submission form on your behalf. 3. Thank you for stating the following in the Acknowledgments Section of your manuscript: "This study was supported by Agricultural Joint Special Project of Yunnan Provincial Department of Science and Technology, grant number 202301BD070001-087; Open Research Program of Key Laboratory of Breeding and Utilization of Resource Insects of National Forestry and Grassland Administration, grant number XXX; Central Finance Forestry Science and Technology Promotion  Demonstration Project, grant number Yun[2024]TG19, and Open project of Key Laboratory for Forest Resources Conservation and Utilization in the Southwest Mountains of China, Ministry of Education, grant number KLESWFU——202005." We note that you have provided funding information that is not currently declared in your Funding Statement. However, funding information should not appear in the Acknowledgments section or other areas of your manuscript. We will only publish funding information present in the Funding Statement section of the online submission form. Please remove any funding-related text from the manuscript and let us know how you would like to update your Funding Statement. Currently, your Funding Statement reads as follows: ""This study was supported by Agricultural Joint Special Project of Yunnan Provincial Department of Science and Technology, grant number 202301BD070001-087; Open Research Program of Key Laboratory of Breeding and Utilization of Resource Insects of National Forestry and Grassland Administration, grant number XXX; Central Finance Forestry Science and Technology Promotion Demonstration Project, grant number Yun[2024]TG19 , and Open project of Key Laboratory for Forest Resources Conservation and Utilization in the Southwest Mountains of China, Ministry of Education, grant number KLESWFU——202005." Please include your amended statements within your cover letter; we will change the online submission form on your behalf. 4. We note that your Data Availability Statement is currently as follows: All relevant data are within the manuscript and its Supporting Information files. Please confirm at this time whether or not your submission contains all raw data required to replicate the results of your study. Authors must share the “minimal data set” for their submission. PLOS defines the minimal data set to consist of the data required to replicate all study findings reported in the article, as well as related metadata and methods (https://journals.plos.org/plosone/s/data-availability#loc-minimal-data-set-definition). For example, authors should submit the following data: - The values behind the means, standard deviations and other measures reported;- The values used to build graphs;- The points extracted from images for analysis. Authors do not need to submit their entire data set if only a portion of the data was used in the reported study. If your submission does not contain these data, please either upload them as Supporting Information files or deposit them to a stable, public repository and provide us with the relevant URLs, DOIs, or accession numbers. For a list of recommended repositories, please see https://journals.plos.org/plosone/s/recommended-repositories. If there are ethical or legal restrictions on sharing a de-identified data set, please explain them in detail (e.g., data contain potentially sensitive information, data are owned by a third-party organization, etc.) and who has imposed them (e.g., an ethics committee). Please also provide contact information for a data access committee, ethics committee, or other institutional body to which data requests may be sent. If data are owned by a third party, please indicate how others may request data access. 5. Please include your full ethics statement in the ‘Methods’ section of your manuscript file. In your statement, please include the full name of the IRB or ethics committee who approved or waived your study, as well as whether or not you obtained informed written or verbal consent. If consent was waived for your study, please include this information in your statement as well. 6. Please include a separate caption for each figure in your manuscript. 7. Please upload a copy of Supporting Information Figure/Table/etc. "Supporting information" which you refer to in your text on page 24.

Reviewers' comments:

Reviewer's Responses to Questions

**Comments to the Author**

1. Is the manuscript technically sound, and do the data support the conclusions?

Reviewer #1: Yes

Reviewer #2: Yes

2. Has the statistical analysis been performed appropriately and rigorously? 

Reviewer #1: Yes

Reviewer #2: Yes

3. Have the authors made all data underlying the findings in their manuscript fully available?

Reviewer #1: Yes

Reviewer #2: Yes

4. Is the manuscript presented in an intelligible fashion and written in standard English?

Reviewer #1: Yes

Reviewer #2: Yes

5. Review Comments to the Author

Reviewer #1: The manuscript submitted by Wang et al. mainly dealt with analysis of the release pattern of floral aroma components of Rhus chinensis based on HS-SPME-GC-MS technique. The article is interesting and applicable. However, some problems or errors should be revised or explained according to the following suggestions.

title: coyunmponents should be components

Abstract

Please add one sentence to introduce the background of this study.

line 105, 80 celcius degree might be very high, why did you select such a high temperature? I worried about the volatiles being degraded.

line 106, please tell us what's the type of the fiber of SPME ?

line 120, relative retention time of each peak, aiming to determine the diverse volatile components in the aroma of R. chinensis flowers corresponding to each peak. why not use RI or kovatts index.

Table 1, RT should be revised into RI

Reviewer #2: The title of this article is "Analysis of the release pattern of floral aroma components of Rhus chinensis based on HS-SPME-GC-MS technique. Headspace-solid phase microextraction was used to identify the essential components of the floral aroma during the budding, blooming, and withering of the flowering stages of Rhus chinensis.. Overall, the article is very interesting, the experiments are properly designed, the data is sufficient, the results are impressive and meaningful. The revision advice is as below:

1，Please add the significance of the study aim in the abstract.

2，Lines 29 and 30: "reaching 3.60μg/g at the full bloom stage 29 and only 2.40μg/g after the bloom stage." Please check for errors, as there is a discrepancy between the current abstract and the results. It is advisable to correct.

3，The introduction is too concise. More literature regarding the research scope is recommended, rather than discussing the scent of honey.

4，The authors report that the floral aroma compounds were identified through each peak's retention time and peak area. However, the total ion current (TIC) chromatograms for the determination of floral aroma compounds are not presented in the results. It is advisable to submit the relevant supplementary materials.

5，Since Table 2 only lists compound classes without specific identifications. This complicates result interpretation. I suggest including a table with the 84 compounds in the Supplementary Materials.

6，Tables should use three-wire tables.

7, Acknowledgments： Project support numbers should be added.

6. PLOS authors have the option to publish the peer review history of their article (what does this mean? ). If published, this will include your full peer review and any attached files.

**Do you want your identity to be public for this peer review?** For information about this choice, including consent withdrawal, please see our Privacy Policy .

Reviewer #1: No

Reviewer #2: **Yes: ** Shu Wang

---

## [Author Response · Author response to Decision Letter 1]

11 Jan 2025

Dear Mozaniel Santana de Oliveira, Ph.D,

Subject: Revised Manuscript - PONE-D-24-52009R1

On behalf of all the contributing authors, I would like to express our sincere appreciation for your time involved in reviewing the manuscript and your constructive comments concerning our article entitled "Analysis of the release pattern of floral aroma coyunmponents of Rhus chinensis based on HS-SPME-GC-MS technique". These comments are all valuable and helpful for improving our article. We have read through the comments carefully and have made corrections. Based on the instructions provided in your comments, we uploaded the file of the revised manuscript. In this revised version, changes to our manuscript were all highlighted within the document by using red-colored text.

In the remainder of this letter, we discuss each of your comments individually along with our corresponding responses. To facilitate this discussion, we first retype your comments in italic font and then present our responses to the comments.

Journal Requirements:

Comment 1: Please ensure that your manuscript meets PLOS ONE's style requirements, including those for file naming. The PLOS ONE style templates can be found at https://journals.plos.org/plosone/s/file?id=wjVg/PLOSOne_formatting_sample_main_body.pdf and https://journals.plos.org/plosone/s/file?id=ba62/PLOSOne_formatting_sample_title_authors_affiliations.pdf

Response 1: Thank you for your comments. In accordance with the downloaded template and the format requirements of your journal, we have meticulously examined and improved the content in the manuscript to render it more compliant with the publication norms and requirements of your journal. However, we haven't found the relevant requirements of your journal for file naming. For this reason, I have sent relevant emails to your journal for help. As of now, I still haven't received a relevant reply. I am very sorry about this.

Comment 2: Thank you for stating the following financial disclosure:

"This study was supported by Agricultural Joint Special Project of Yunnan Provincial Department of Science and Technology, grant number 202301BD070001-087; Open Research Program of Key Laboratory of Breeding and Utilization of Resource Insects of National Forestry and Grassland Administration, grant number XXX; Central Finance Forestry Science and Technology Promotion Demonstration Project, grant number Yun[2024]TG19 , and Open project of Key Laboratory for Forest Resources Conservation and Utilization in the Southwest Mountains of China, Ministry of Education, grant number KLESWFU——202005."

Response 2: Thanks for your careful checks. The relevant funders and other authors in this study were all involved, and their roles and contributions in this study are as follows(page 22):

Author Contributions:

Conceptualization: Ju Gu, Chao Wang.

Methodology: Ju Gu.

Software: Yun Niu.

Validation: Ju Gu, Yan-di Wu, Yi-ting Tang.

Formal analysis: Ju Gu.

Investigation: Ping Liu, Yi-ting Tang.

Resources: Chao Wang, Zi-xiang Yang.

Data curation: Ju Gu.

Writing—original draft preparation: Ju Gu.

Writing—review and editing: Ju Gu.

Visualization, Yan-di Wu.

Supervision: Zi-xiang Yang, Chao Wang.

Project administration, Chao Wang.

Funding acquisition, Chao Wang, Ping Liu, Zi-xiang Yang.

All authors have read and agreed to the published version of the manuscript.

Comment 3: Thank you for stating the following in the Acknowledgments Section of your manuscript:

"This study was supported by Agricultural Joint Special Project of Yunnan Provincial Department of Science and Technology, grant number 202301BD070001-087; Open Research Program of Key Laboratory of Breeding and Utilization of Resource Insects of National Forestry and Grassland Administration, grant number XXX; Central Finance Forestry Science and Technology Promotion Demonstration Project, grant number Yun[2024]TG19, and Open project of Key Laboratory for Forest Resources Conservation and Utilization in the Southwest Mountains of China, Ministry of Education, grant number KLESWFU——202005."

""This study was supported by Agricultural Joint Special Project of Yunnan Provincial Department of Science and Technology, grant number 202301BD070001-087; Open Research Program of Key Laboratory of Breeding and Utilization of Resource Insects of National Forestry and Grassland Administration, grant number XXX; Central Finance Forestry Science and Technology Promotion Demonstration Project, grant number Yun[2024]TG19 , and Open project of Key Laboratory for Forest Resources Conservation and Utilization in the Southwest Mountains of China, Ministry of Education, grant number KLESWFU——202005."

Response 3: We are extremely grateful for this opinion. In order to effectively address this issue, we have removed the text related to the funding information contained in the manuscript according to the opinion. The content of the funding particulars that need to be updated is altered as follows:

This study was supported by Agricultural Joint Special Project of Yunnan Provincial Department of Science and Technology, grant number 202301BD070001-087; National Natural Science Foundation of China, grant number 32470544; Central Finance Forestry Science and Technology Promotion Demonstration Project, grant number Yun[2024]TG19 , and Open project of Key Laboratory for Forest Resources Conservation and Utilization in the Southwest Mountains of China, Ministry of Education, grant number KLESWFU——202005.

Comment 4: We note that your Data Availability Statement is currently as follows: All relevant data are within the manuscript and its Supporting Information files.

Response 4: According to the review comments and their requirements, all the data used in the research results have been uploaded in the Supporting Information files.

Comment 5: Please include your full ethics statement in the ‘Methods’ section of your manuscript file. In your statement, please include the full name of the IRB or ethics committee who approved or waived your study, as well as whether or not you obtained informed written or verbal consent. If consent was waived for your study, please include this information in your statement as well.•Response: According to the review comments and their requirements, all the data used in the research results have been uploaded in the Supporting Information files.

Response 5: We are extremely grateful for your indication of this crucial point. The plant samples utilized in this research were collected from the Institute of Plateau Forestry, Chinese Academy of Forestry Sciences. Prior to the commencement of the experiment, permission from this institution has been obtained. The ethics statement has been added to the “Experimental materials” section within “Materials and methods”(Materials and methods, Line 100-103, Page 5).

Comment 6: Please include a separate caption for each figure in your manuscript.

Response 6: Thanks for your careful checks. Based on your comments, we have included a separate caption for each figure after the position where the picture is cited in the manuscript.

Comment 7: Please upload a copy of Supporting Information Figure/Table/etc. "Supporting information" which you refer to in your text on page 24.

Response 7: Thanks for your careful checks. We have completely uploaded the figures/tables and other files in the supplementary information mentioned on page 24 of the main text.

Reviewer 1

Comment 1: title: coyunmponents should be components

Response 1: Thank you for pointing this out. We apologize for our carelessness. Based on your comment, we have corrected the "coyunmponents" into "components"(Titie, Line 4, Page 1).

Comment 2: Abstract: Please add one sentence to introduce the background of this study.

Response 2: We appreciate this comment. To address this concern, we have added the purpose and significance of the research in the first sentence of the abstract: “Rhus chinensis, a native plant species of China, possesses significant economic value in the ornamental sector. This study investigates the floral fragrance components and release patterns of R. chinensis, thus providing a theoretical foundation for the utilization of its floral fragrance.”(Abstract, Line 24-27, Page 1).

Comment 3: line 105, 80 celcius degree might be very high, why did you select such a high temperature? I worried about the volatiles being degraded.

Response 3: Thank you for your comments, the discussion regarding this question is presented following: In this study, the temperature is optimally selected. Most of the aroma components in plant flowers are volatile organic compounds (VOCs). At normal temperature, the volatilization rate of these compounds is relatively slow. When heated to 80 degrees Celsius, the molecular movement is accelerated, making the volatile compounds easier to be released and enter the headspace area. For example, common terpenes, esters, etc. in flowers have limited volatility at normal temperature, but at 80 degrees Celsius, they can volatilize and be extracted well, and at the same time, there will be no large amount of decomposition phenomenon, providing sufficient target substances for subsequent solid-phase microextraction.

Comment 4: line 106, please tell us what's the type of the fiber of SPME ?

Response 4: Thanks for your careful reading. We apologize for our carelessness. In this study, the fiber type used in solid-phase microextraction (SPME) is 75 μm CAR/PDMS and we have added and refined in the HS-SPME analysis of Materials and Methods(Materials and Methods, Line 123, Page 6).

Comment 5: line 120, relative retention time of each peak, aiming to determine the diverse volatile components in the aroma of R. chinensis flowers corresponding to each peak. why not use RI or kovatts index.

Response 5: Thank you for pointing this out. In this study, floral fragrance components were determined by gas chromatography-mass spectrometry (GC-MS). The analysis was conducted by combining the mass spectrometry database and the relative retention time of each peak. Subsequently, under the same conditions, n-alkane standards (C7-C20) were used as references to calculate the linear retention indices of different components. After comparison with the NIST online database, the components were confirmed. This has been improved in “Data Analysis”(Data Analysis, Line 139-143, Page 6-7).

Comment 6: Table 1, RT should be revised into RI

Response 6: Thank you for underlining this deficiency. This section was revised and modified according to the suggestions shown by you. The “RT” in subsequent tables has been replaced by the retention index “RI”(Table 1, Page 9-11).

Reviewer 2

Comment 1: Please add the significance of the study aim in the abstract.

Response 1: Thanks for your careful reading. Based on your comment, the purpose and significance of the research have been added in the first sentence of the abstract (Abstract, Line 24-27, Page 1).

Comment 2: Lines 29 and 30: "reaching 3.60μg/g at the full bloom stage 29 and only 2.40μg/g after the bloom stage." Please check for errors, as there is a discrepancy between the current abstract and the results. It is advisable to correct.

Response 2: Thanks for your careful reading. Have checked and revised and completed. Changes Made: Abstract: "reaching 3.60μg/g at the full bloom stage 29 and only 2.40μg/g after the bloom stage." has been changed to "reaching 3.60μg/g at the full flowering stage and only 2.40μg/g after the withering stage."(Abstract, Line 41-42, Page 2).

Comment 3: The introduction is too concise. More literature regarding the research scope is recommended, rather than discussing the scent of honey.

Response 3: We think this is an excellent suggestion. We have deleted the descriptions about honey in the manuscript and added the descriptions about the combination of GC-MS and multivariate analysis such as PCA(Introduction, Line 54-59, Line 81, Page 3-4).

Comment 4: The authors report that the floral aroma compounds were identified through each peak's retention time and peak area. However, the total ion current (TIC) chromatograms for the determination of floral aroma compounds are not presented in the results. It is advisable to submit the relevant supplementary materials.

Response 4: Thanks for your careful checks. Based on your comments, we have added the Total ion chromatogram of floral fragrance components of R. chinensis in the Supporting information(S1 Fig, S2 Fig).

Comment 5: Since Table 2 only lists compound classes without specific identifications. This complicates result interpretation. I suggest including a table with the 84 compounds in the Supplementary Materials.

Response 5: Thanks for your careful checks. Based on your comments, we have added a table with the 84 compounds in the Supplementary Materials(S1 Table).

Comment 6: Tables should use three-wire tables.

Response 6: Thank you very much for your comments. It seems that the table format requirements given by your journal are not for three-wire tables. I'm wondering whether it's necessary to change it into a three - line table.

Comment 7: Acknowledgments: Project support numbers should be added.

Response 7: We sincerely appreciate the valuable comments. According to your suggestions, the project numbers have been perfected. In line with Comment 2 in the previous journal requirements, we updated the project and deleted(Page 22) the text related to the funding information included in the manuscript.

The funded project and its number have been updated to:

This study was supported by Agricultural Joint Specia

---

## [Decision Letter · Decision Letter 1]

29 Jan 2025

Analysis of the release pattern of floral aroma components of Rhus chinensis based on HS-SPME-GC-MS technique

PONE-D-24-52009R1

Dear Dr. Wang%,

We’re pleased to inform you that your manuscript has been judged scientifically suitable for publication and will be formally accepted for publication once it meets all outstanding technical requirements.

Kind regards,

Mozaniel Santana de Oliveira, Ph.D

Academic Editor

PLOS ONE

Additional Editor Comments (optional):

Reviewers' comments:

Reviewer's Responses to Questions

**Comments to the Author**

1. If the authors have adequately addressed your comments raised in a previous round of review and you feel that this manuscript is now acceptable for publication, you may indicate that here to bypass the “Comments to the Author” section, enter your conflict of interest statement in the “Confidential to Editor” section, and submit your "Accept" recommendation.

Reviewer #2: All comments have been addressed

2. Is the manuscript technically sound, and do the data support the conclusions?

Reviewer #2: Yes

3. Has the statistical analysis been performed appropriately and rigorously? 

Reviewer #2: Yes

4. Have the authors made all data underlying the findings in their manuscript fully available?

Reviewer #2: Yes

5. Is the manuscript presented in an intelligible fashion and written in standard English?

Reviewer #2: Yes

6. Review Comments to the Author

Reviewer #2: The author has answered my questions and revised the paper carefully. I have no additional suggestions at this time.

7. PLOS authors have the option to publish the peer review history of their article (what does this mean? ). If published, this will include your full peer review and any attached files.

**Do you want your identity to be public for this peer review?** For information about this choice, including consent withdrawal, please see our Privacy Policy .

Reviewer #2: No

---

## [Editor Report · Acceptance letter]

PONE-D-24-52009R1

PLOS ONE

Dear Dr. Wang,

I'm pleased to inform you that your manuscript has been deemed suitable for publication in PLOS ONE. Congratulations! Your manuscript is now being handed over to our production team.

Kind regards,

on behalf of

Dr. Mozaniel Santana de Oliveira

Academic Editor

PLOS ONE